# The Promotion and Development of One Health at Swiss TPH and Its Greater Potential

**DOI:** 10.3390/diseases10030065

**Published:** 2022-09-14

**Authors:** Jakob Zinsstag, Karin Hediger, Yahya Maidane Osman, Said Abukhattab, Lisa Crump, Andrea Kaiser-Grolimund, Stephanie Mauti, Ayman Ahmed, Jan Hattendorf, Bassirou Bonfoh, Kathrin Heitz-Tokpa, Mónica Berger González, Alvar Bucher, Monique Lechenne, Rea Tschopp, Brigit Obrist, Kristina Pelikan

**Affiliations:** 1Swiss Tropical and Public Health Institute, CH-4123 Allschwil, Switzerland; 2University of Basel, CH-4003 Basel, Switzerland; 3Jigjiga University, Jigjiga P.O. Box 1020, Ethiopia; 4Institute of Endemic Diseases, University of Khartoum, Khartoum 11111, Sudan; 5Sudanese National Academy of Sciences, Khartoum 11111, Sudan; 6Centre Suisse de Recherches Scientifiques, Abidjan 01 BP 1303, Côte d’Ivoire; 7Universitdad del Valle, Unidad de Antropologia Médica, Guatemala City 01015, Guatemala; 8Armauer Hansen Research Institute, Addis Ababa 1005, Ethiopia

**Keywords:** One Health, history of ideas, theory, ontology, epistemology, methods, transdisciplinary, rabies, brucellosis, surveillance-response, language

## Abstract

One Health, an integrated health concept, is now an integral part of health research and development. One Health overlaps with other integrated approaches to health such as EcoHealth or Planetary Health, which not only consider the patient or population groups but include them in the social-ecological context. One Health has gained the widest foothold politically, institutionally, and in operational implementation. Increasingly, One Health is becoming part of reporting under the International Health Legislation (IHR 2005). The Swiss Tropical and Public Health Institute (Swiss TPH) has played a part in these developments with one of the first mentions of One Health in the biomedical literature. Here, we summarise the history of ideas and processes that led to the development of One Health research and development at the Swiss TPH, clarify its theoretical and methodological foundations, and explore its larger societal potential as an integrated approach to thinking. The history of ideas and processes leading to the development of One Health research at the Swiss TPH were inspired by far-sighted and open ideas of the directors and heads of departments, without exerting too much influence. They followed the progressing work and supported it with further ideas. These in turn were taken up and further developed by a growing number of individual scientists. These ideas were related to other strands of knowledge from economics, molecular biology, anthropology, sociology, theology, and linguistics. We endeavour to relate Western biomedical forms of knowledge generation with other forms, such as Mayan medicine. One Health, in its present form, has been influenced by African mobile pastoralists’ integrated thinking that have been taken up into Western epistemologies. The intercultural nature of global and regional One Health approaches will inevitably undergo further scrutiny of successful ways fostering inter-epistemic interaction. Now theoretically well grounded, the One Health approach of seeking benefits for all through better and more equitable cooperation can clearly be applied to engagement in solving major societal problems such as social inequality, animal protection and welfare, environmental protection, climate change mitigation, biodiversity conservation, and conflict transformation.

## 1. Introduction

One Health, an integrated health concept, is now an integral part of health research and development. One Health overlaps with other integrated approaches to health such as EcoHealth [1,2] or Planetary Health [3,4], which not only consider the patient or population groups, but include them in the social-ecological context [5]. Integrated approaches are warranted today to address the complex problems that cannot be solved by single disciplines such as climate change, biodiversity loss, luring antimicrobial resistance, but also inequality, discrimination, government effectiveness, and corruptions, who all have a heavy toll on the public’s health. One Health has gained the widest foothold politically, institutionally and in operational implementation. Increasingly, One Health is becoming part of reporting under the International Health Regulations (IHR 2005) [6]. In the Carbis Bay Declaration of 12 July 2021, the group of the seven largest industrialized countries (G7) committed itself to promote One Health and wants to incorporate One Health into all policy areas [7]. The Swiss Tropical and Public Health Institute (Swiss TPH) has played a part in these developments with one of the first mentions of One Health in the biomedical literature [8,9]. In this paper, we summarize the history of ideas and processes that led to the development of One Health research and development at the Swiss TPH, clarify its theoretical and methodological foundations, and explore its larger societal potential as an integrated approach to thinking.

## 2. Calvin Schwabe’s One Medicine

The first author first came into contact with “One Medicine” the precursor concept of One Health in 1985, when his doctoral supervisor Hans Fey at the Veterinary Faculty of the University of Bern showed him Calvin Schwabe’s book: “Veterinary Medicine and Human Health”. Calvin Schwabe postulated that there is no paradigmatic difference between human and veterinary medicine, that both sciences have the same foundations in anatomy, physiology, and the origin of diseases [10]. Schwabe, an American veterinary epidemiologist who taught at the American University of Beirut, drew inspiration from Dinka healers in what is today South Sudan. The Dinka people lived almost exclusively depending on animal husbandry and wandered with their animals in search of food and water. Healers took care of the health of people and animals [11]. Schwabe was impressed by the integrated way of life of the Dinka, which included all areas of human life, and contrasted it with the reductionist, rationalist-empiricist way of modern scientific thinking [12]. Schwabe’s modern scientific thinking was influenced by the integrated approach of the Dinka. An integrated approach to health research that involves humans, animals, and their environment simultaneously becomes systemic and seeks to incorporate the complex interactions of human-environment systems. This requires a theoretical basis to develop methods and case studies. The postulate of One Medicine, the paradigmatic unity of human and veterinary medicine, thus has African roots.

## 3. Marcel Tanner’s Impulse

In 1996, Marcel Tanner, then director of the Swiss Tropical Institute (STI), visited the Centre Suisse de Recherches Scientifiques in Abidjan, Côte d’Ivoire, and suggested to Jakob Zinsstag, a veterinarian specializing in parasitology, that he look into the health of mobile livestock owners in Chad. During this time, the STI had been active in Chad for some time, working on improvements to health care in the Chari-Baguirmi and Lake Chad region. It became apparent that the mobile livestock keepers (pastoral nomads) could not be reached with the existing health care system. The existing system of stationary health centres was not adapted to the mobile lifestyle of mobile livestock keepers. Tanner’s far-reaching idea to entrust a veterinarian with the study of health care for pastoral nomads stemmed from his taking a more expansive view of health and well-being embedded in a broader socio-ecological context. For Tanner, veterinarians seemed to be the ideal animators and catalysts of this process. At that time, Zinsstag already had contacts with Idriss Oumar Alfaroukh, the director of the Laboratoire de Recherches Vétérinaires et de Zootechnie (LRVZ) in Farcha, N’Djaména, Chad. A grant from the Swiss National Science Foundation (SNSF) launched a long-term research partnership based on Schwabe’s One Medicine as a starting point.

## 4. «Santé des Nomades» in Chad

In 1998, the “Santé des Nomades” project began in Chad with the first doctoral students, Colette Djaibe-Diguimbaye and Esther Schelling, which built on the principles for research partnerships of the Swiss Academies of Sciences [13], establishing repeated direct contact with the nomadic population and the authorities [14]. At the same time, an anthropologist and a geographer, supervised by Kaspar Wyss and Marcel Tanner, investigated the influence of cultural conditions and geographical mobility on health care. A team of human medical personnel, veterinarians, and microbiologists simultaneously examined the health of people and their livestock, including cows, sheep, goats, camels, horses, and donkeys. It turned out that more livestock were regularly vaccinated than children. Not a single child was fully vaccinated against the usual childhood diseases. In further participatory meetings with the population and the authorities, the stakeholders agreed to offer joint vaccination services for animals and humans at the same time. When veterinarians organized vaccination campaigns against anthrax (*Bacillus anthracis*) or Contagious Bovine Pleuropneumonia (CBPP), they were accompanied by human medical personnel, who could vaccinate children and women, conduct health education, and dispense medicine at the same time [15,16]. Thanks to the shared transport and cold chain, there was a 15% cost savings compared to separate health services [16]. For the first time, an added value of closer cooperation between human and veterinary medicine in the field of health care research could be demonstrated. This laid the foundation for an inductive, theoretical and methodological extension of One Medicine to One Health: joint studies in humans and animals and transdisciplinary, participatory processes as a basis for closer collaboration.

## 5. Brucellosis Control in Mongolia

After the end of the socialist period in the early 1990s, brucellosis flared up again rapidly in Mongolia because health services were no longer adequately funded. International experts recommended that the World Health Organization (WHO) reintroduce a mass vaccination of livestock to control brucellosis in humans. WHO reached the STI with a question: Is it worth vaccinating 25 million livestock (sheep, goats, cows, yaks, and camels) in Mongolia against brucellosis to protect humans? Health economist Felix Roth approached the veterinarian Jakob Zinsstag to contribute expertise on clinical brucellosis in livestock. The question of how an intervention in veterinary medicine would affect public health was difficult to assess and could not be answered using statistical methods. Mathematical statistician Penelope Vounatsou was instrumental in writing a mathematical model, the first animal–human brucellosis transmission model [17]. The central part of the model describes the change in the number of diseased humans I_h_ as a function of the number of contagious animals I_a_ and susceptible humans S_h_ (Equation (1)):(1)$$\frac{dI_{h}}{dt}=\beta I_{a}S_{h}$$

This mathematical model was used to simulate the effect of animal vaccination on human health. The related inter-sectoral economic analysis showed that mass vaccination of animals was not profitable for public health alone. However, when the gains for private health costs, prevented income loss and increased animal production were added, the intervention became profitable with a benefit-cost ratio of three to one (Equation (2)) [18].
(2)$$\frac{Public\;health\;and\;animal\;benefits}{Intervention\;cost\;in\;livestock}>\frac{Public\;health\;benefits}{Intervention\;cost\;in\;livestock}\;\;$$

Thus, a methodological extension to dynamic animal-human systems was achieved and an added value of a closer cooperation between human and veterinary medicine was once again demonstrated, even taking into account climate phenomena, through consideration of the additional mortality of animals due to snowstorm catastrophes. Only the unique constellation of the STI, with in this case, economic, veterinary, and mathematical-statistical expertise, made this breakthrough possible, although Marcel Tanner was sceptical at the beginning.

## 6. From One Medicine to One Health

Through the National Centre of Competence in Research (NCCR North-South, www.nccr-north-south.ch, accessed on 9 August 2022 ), jointly funded by the SNSF and the Swiss Agency for Development and Cooperation (SDC), integrated research on health was decisively strengthened and expanded. The NCCR North-South promoted inter- and transdisciplinary research approaches, to which the research at the STI in turn contributed [19]. Thanks to Marcel Tanner, contact was made with Brigit Obrist, a medical anthropologist at the University of Basel and the STI, and the work with pastoral nomads was extended to Mauritania and Mali. In discussions with Brigit Obrist and Marcel Tanner, an exchange arose on the concept of “access to health care” [20] which was expanded to the concept of “equity effectiveness” [21] and led to a quantifiable Equation (3) [22] embedded methodologically in a mixed qualitative-quantitative approach [23].
(3)$$Effectiveness=\alpha \overset{n}{\underset{i=1}{\prod }}\beta_{\left(i\right)}$$

For interventions, especially at the animal-human interface, to be fully effective, a sequence of multiplicatively linked access criteria such as affordability or adequacy (*β*) must be met. The access criteria are further multiplied with the efficacy α of a drug or a vaccine. This requires a participatory, transdisciplinary dialogue, in which stakeholders’ population groups, authorities, and health care providers agree on the best way to implement interventions. One Health research and implementation is clearly at the interface between qualitative and quantitative methods, with far reaching consequences (see below One Health paradigms and ontology).

## 7. Rabies: The Added Value of a One Health Approach

One Health research is strongly partnership-based. Whenever possible, researchers are trained in partner countries. PhD students become future project partners in national authorities and research institutes after completing their doctorate. Well-trained project partners strengthen mutual trust and intensify dialogue around new research topics. For example, Kebkiba Bidjeh, then the director of the LRVZ in Chad, proposed to study rabies control. At that time, rabies, a viral disease without its own laboratory capacity, was not at the forefront of STI’s interest. Nonetheless, it was possible to establish a collaboration with the Swiss Rabies Centre in Bern and establish the immunofluorescence diagnosis of rabies at the LRVZ. Initial studies estimated an incidence of 1.4 rabid dogs per thousand per year in the urban area of N’Djaména [24]. A first attempt to mass vaccinate 3000 dogs in N’Djaména demonstrated that 70% of the dogs could be vaccinated and that only about 15% of the dogs were ownerless and thus not accessible for vaccination [25]. When the Chadian Minister of Health was asked about the mass vaccination of dogs to eliminate human rabies, he replied that he was responsible for the people and not for the dogs. Asked about the same issue, the Minister of Animal Production replied that his priority were cattle and not dogs. The narrow vision of the government sector was the main barrier that we faced to starting a mass vaccination of dogs to eliminate rabies in humans and led us to ask: What costs less, the post-exposure prophylaxis (PEP) of humans after the bite of a rabies suspected animal or the mass vaccination of dogs along with simultaneous human PEP? This question led to the development of the first mathematical dog–human rabies transmission model. The associated economic analysis demonstrated that over a period of ten years, the cumulative cost to humans of mass vaccination of dogs with PEP was less than PEP alone, because vaccination of dogs can interrupt the transmission of rabies. For the control and elimination of rabies in dogs, the One Health approach clearly adds value (Equation (4)).
Cumulative cost _(Dog mass vaccination + PEP)_ < Cumulative cost _(PEP)_(4)

Although we did not initially want to work on rabies, this partnership with the LRVZ in Chad became one of the most successful One Health projects with far-reaching methodological extensions. Research on rabies was expanded into Côte d’Ivoire and Mali, where blockchain secured the electronic patient registration for rabies PEP with canine diagnostics and vaccine supply chain starts to be tested in 2022. With this first blockchain project at Swiss TPH, One Health is contributing to the digitisation of the interfaces between human and veterinary medicine and pharmaceutical care.

## 8. Creation of a One-Health Unit in Ethiopia

In 2005, Rea Tschopp travelled to Ethiopia to start her PhD on the epidemiology of Bovine tuberculosis at the animal–human interface. This initial work led to a long-standing fruitful collaboration between Swiss TPH and Armauer Hansen Research Institute (AHRI) lasting to this date in the form of a joint research appointment. Over the next 16 years, Rea Tschopp developed a One-Health team at AHRI, training Ethiopian staff in One-Health, increasing awareness of the One-Health concept at national level through academic teaching and the engagement into the discussion of a possible national One-Health platform in 2011 with supportive partners such as USAID. This resulted in the signature of an MoU in 2016 between four key Ministries in Ethiopia to create a National One-Health Steering Committee. The One-Health team at AHRI expanded over the years under the leadership of Rea Tschopp, in terms of staff and projects, and the Ethiopia Government ratified it as a One-Health Unit in 2016. The unwavering support and vision of Marcel Tanner over the years was essential in strengthening this collaboration.

## 9. The Jigjiga University One Health Initiative

Building on the One Health unit at AHRI, the Swiss Agency for Development and Cooperation (SDC) committed to the development of a partnership with Jigjiga University in the Jigjiga University One Health Initiative (JOHI). JOHI builds up academic and development capacity in One Health to better serve the health and livelihoods of pastoralists and their animals in the Somali Region of Ethiopia. The partnership is further extended to Somaliland. Among others, integrated Surveillance-Response Systems (iSRS) became an important theme in JOHI. In Adadle district in the Somali region of Ethiopia, an iSRS was developed involving village communities, community workers, and local health and veterinary authorities, thereby achieving a high degree of surveillance sensitivity. Rapid linkage with a molecular diagnostic laboratory remains a challenge, but the system has the potential to achieve very fast etiological elucidation of new disease outbreaks in humans and animals [26].

## 10. From Communicable to Non-Communicable Diseases

In 2009, the Institute of Social and Preventive Medicine at the University of Basel and the STI merged to form the Swiss Tropical and Public Health Institute (Swiss TPH). This expanded the area of competence of the new Swiss TPH to include non-contagious diseases, cancer epidemiology, and environmental exposure research. A first exchange on One Health took place with Charlotte Braun-Fahrländer’s group on the relationship between exposure to a farm environment and the incidence of asthma. Exposure to animals on farms has been shown to lead to a lower risk of asthma. With Nicole Probst-Hensch, we investigated the possibilities of joint surveillance for cancer in animals and humans. Since dogs age faster and therefore develop cancer more quickly, it is quite conceivable that they could have important sentinel functions for humans.

During this time, we also worked with Andrea Meisser, who established contact with Karin Hediger, a psychologist working with animal-assisted therapy. A very fruitful collaboration developed with the Clinic for Neurorehabilitation and Paraplegiology (REHAB) in Basel and the institute for interdisciplinary research on human animal relationship (IEMT), to which Swiss TPH provided an institutional home. From a One Health perspective, in animal-assisted therapy it must be ensured that humans are not cared for at the detriment of animals. To avoid this danger, it is important to monitor the health and well-being of the animals during such therapies. Here too, the aim is to achieve added value in health for humans and animals [27,28].

## 11. From the Textbook to the Online Course and the One Health Platform

Existing international networks at Swiss TPH became accessible to One Health. For example, Marcel Tanner initiated contact with Maxine Whittaker at the University of Queensland (UQ) in Brisbane, Australia. After Joint international courses on One Health between UQ and Swiss TPH, Maxine Whittaker proposed writing a textbook on One Health, the first edition of which was published in 2015 [29]. In 2020, a French translation was published and a second English edition was completed [30,31]. In 2022, a Mandarin translation will be published under the direction of Guojing Yang from the Hainan Medical University.

During the same time, we thought about how One Health education could have a greater impact. At the request of the students at the Swiss TPH, we expanded the introductory course to One Health to include an advanced course, where students work independently on mathematical models and economic analyses and learn the basics of transdisciplinary methods. Following a proposal from the University of Basel, we developed a Massive Open Access Online Course (MOOC), with the New Media Centre, the first such course at Swiss TPH (www.futurelearn.com/courses/one-health, accessed on 9 August 2022). In seven tutored editions of the 6-week course offered through 2021, we reached more than 10,000 learners worldwide, achieving global leverage for One Health education. In the textbook and online course, we set out the basic principles and a definition of One Health (Box 1).

Box 1Foundational principles of One Health adapted with permission from [30].One Health is about cooperation between different academic disciplines underlying human and veterinary medicine in the first place, but without any barrier to natural and social sciences and the humanities. One Health also engages with non-academic actors in the co-production of knowledge.Cooperating partners will seek a benefit of working together sooner or later. To fully understand the range of potential benefits of a closer cooperation implies a deeper and comprehensive recognition and understanding of how humans and animals and their environment are interrelated at all scales. This is a necessary requirement of One Health but still not sufficient.A sufficient requirement for One Health is demonstrating the benefits and added values resulting from the crosstalk and closer cooperation between human and animal health and all related disciplines and stakeholders.We therefore define One Health as any added value in terms of improved health and wellbeing of humans and animals, financial savings, social resilience and environmental sustainability achievable by the cooperation of human and veterinary medicine and other disciplines when compared to the two medicines and other disciplines working separately.

CABI, the publisher of our textbook, entrusted Jakob Zinsstag and Lisa Crump as editors in developing CABI One Health Resources, beginning in 2021. The platform consists of an online scientific journal, a knowledge bank, and a One Health case study database (www.cabi.org/products-and-services/one-health-resources-cabi, accessed on 9 August 2022 ).

## 12. One Health Parasitology with Jürg Utzinger

From 2015, the new director of Swiss TPH Jürg Utzinger continued to promote the development of One Health in parasitology projects. In a study on liver flukes (*Fasciola* spp.) and schistosomiasis (*Schistosoma* spp.) in the Lake Chad area, both diseases were found in animals and schistosomiasis was identified in humans. The most effective drugs against these diseases, praziquantel and triclabendazole, are not available for humans or animals in Chad. An in-depth genetic study in Côte d’Ivoire found hybrid forms of *Schistosoma haematobium* and *Schistosoma bovis* [32]. This observation had implications for drug efficacy, indicating that human and bovine schistosomes occur in the same habitat and genetic exchange occurs. During the onset of the COVID-19 pandemic in 2020, Jürg Utzinger brought us into contact with Xianong Zhou from the National Institute of Parasitic Diseases and Guojing Yang from Hainan Medical University. This led to an exchange on the progress of building One Health in China and the development of integrated surveillance-response systems [33].

## 13. One Health Paradigms and Ontology

The development of how we think about health, as described above, demonstrates how the health of humans and animals and their environment are inescapably linked. The environment includes the plant and animal biodiversity of ecosystems, but also the plants and livestock and their health. When we try to understand health more comprehensively, our worldview changes. It may move us away from a purely human-centred view towards a multispecies reality that includes the health of humans, their environment and animals, as well. Such a view allows for taking into account human-animal relations in their shared environment [34]. It also moves us away from a completely positivist worldview that assumes an independent quantifiable reality towards a view that assumes multiple, socially influenced realities. This requires a multi-epistemic approach to obtain the most added value out of this collaboration. Inspired by such a multiplicity of ontological realities, we may then learn to critically examine predefined categories and boundaries between humans and animals in interaction with nature and culture. Using a One Health perspective, it becomes possible to assume several complementary worldviews (paradigms) [35]. This makes it easier to do justice to the very different backgrounds and various diverse contexts in which we collaborate. This is particularly evident in projects where communication occurs in multiple languages, representing even more varied mental models of health-illness understandings and their associated values.

## 14. One Health Epistemology

One Health, as the added value [30] of closer cooperation between human and veterinary medicine and other disciplines (Box 1), can be demonstrated with quantitative statistical, mathematical, and economic methods, if we include animal health and other academic disciplines and practical knowledge in addition to human health. When we work in different cultural and linguistic contexts and involve population groups, authorities, and other actors in the research process, we gain additional insights that cannot be captured in numbers and quantities, but which are essential for gaining knowledge, addressing local priorities, and solving societal problems. One Health approaches are most successful when they are open to multiple modes of knowledge generation and receptive to unexpected, emergent outcomes. One Health is multi-epistemic and embraces the thinking in processes as proposed by Alfred North Whitehead as a process philosophy [36] and which also appears in newer One Health evaluation methods [37,38].

## 15. Transdisciplinarity and Multilingual Collaborations

In a transdisciplinary process, all stakeholders bring their experience and knowledge from practice and research into the negotiation. This can result in new so-called implementation knowledge, which combines scientific and knowledge from practice [39]. Thanks to the deepened dialogue between epidemiology and anthropology, the positivist One Health approach expanded to include a constructivist perspective on the way to a multi-epistemic paradigm. This explicitly includes non-academic actors in the creation of knowledge. One Health means that different disciplines and societal actors cooperate with each other (Box 1). In order to find solutions for society as a whole, One Health therefore has a strong transdisciplinary methodological orientation, which was promoted by the cooperation in the NCCR North-South [19]. The research clearly moved in the direction of a public health approach to which the term “One Medicine”, with its clinical connotation, no longer did justice. Marcel Tanner then motivated an analysis of the potential of closer cooperation between human and veterinary medicine to strengthen health systems. In this work, the previous results were summarized and a transition from the concept of “One Medicine” to “One Health”, with a stronger orientation towards public health and health systems, was proposed [9]. In this paper, the term “One Health” was used for the first time in bio-medical literature [8].

## 16. Dialogue between Western Medicine and Maya Indigenous Healers in Guatemala

In Guatemala, in a partnership with the Universidad del Valle and the Releb’aal Saq’e’ Council of Maya Spiritual Guides involving also a telecommunications company, we set up an integrated surveillance-response system for zoonotic diseases using cell phone-based messaging [40]. In this project, a profound dialogue between Maya indigenous healers and Western-trained doctors and veterinarians was established. This culturally diverse, multilingual process was accompanied by anthropologists and linguists. A highlight were case studies in which sick people and animals were simultaneously examined in Maya and Western medicine. The inter-epistemic dialogues overcame a long-standing societal boundary and shed light on how knowledge is gained in both medical systems. The participants agreed to put the patient and the owner of the sick animal in the foreground so that they could decide which forms of treatment they preferred. Unsurprisingly, in all cases individuals decided to adopt solutions from both medical systems, naturally integrating their options. Consistent One Health thinking is thus multiepistemic, necessarily including non-Western knowledge in its attempt to become pluriepistemic (fostering respectful relations) and even inter-epistemic (fostering mutual learning). The multilingual dialogue in Maya Q’eqchi’, Spanish, English, and German pointed to the importance of careful translation and mindful communication to take into account power imbalances and gender differences alongside different knowledge systems.

The transdisciplinary collaboration involved in One Health is only successful if all participants agree on commonly used values and a basic understanding of shared concepts. Since values and concepts manifest themselves in language, the desire for a common language quickly arises in transdisciplinary projects: English is often used as the lingua franca. However, this illusion of a common language causes various difficulties in transdisciplinary projects without automatically leading to successful communication.

Not all projects can use a single language as a lingua franca, because not all project participants speak a common language. Sometimes, several individual languages have to be used as linguae franca, making additional translations necessary. One might see the relevance of this complexity when a concept has no equivalent translation to another, such as the term ‘epidemiology’ or ‘vaccine’ to indigenous languages and dialects, or the Maya term kawilal tz’ultaq’a’ to any latin-germanic language (roughly meaning the wellbeing-health and intact resilience of a live consciousness equivalent to Mother Earth).

In addition, different technical languages and other varieties must be assumed: Each individual language (for example, English) has diverse linguistic characteristics that can be described as varieties. These varieties include the technical languages of all disciplines involved and also of the practice domains. The transition between technical and general language is fluid. Within the individual varieties, different linguistically manifested degrees of expertise can be identified, which do not have to be based on scientific disciplines. When a lingua franca is used, the terms from the corresponding specialist fields are usually retained, and these can cause difficulties in understanding. A project-specific shared technical vocabulary can considerably improve comprehensibility [41].

It is not individual terms that are translated but concepts that are embedded in various communicative practices, and strategies and meaning may be further shaped by local contexts. For correct translations of the individual communication areas, all concepts, communicative practices, and strategies must be known, which is often difficult to achieve. Translations are often carried out by the project staff or by externally hired translators who are not familiar with all the relevant underlying concepts. Mitchell Weiss brought to our attention the concepts of cultural epidemiology emphasizing the awareness for relating an external (etic) and internal (emic) perspective [42]. In intercultural transdisciplinary approaches aiming to add tangible value to One Health goals, a larger degree of success is achieved when time is afforded to developing a preliminary understanding of mental models, constructs, and emic categories of meaning and value of key group participants, as only then successful interpretation and translation between languages (and epistemic systems) can occur. In an intercultural dialogue with Fulani pastoralists in Chad, we quickly found consensus on the understanding of names of animal diseases between Fulani speaking pastoralists and French speaking veterinarians. It was much more difficult to find consensus on human diseases, as local illness perception and Western disease concepts often diverged. This could be because for human diseases and illnesses there was an important difference of perception between the external and internal perspective, whereas for animal diseases, the pastoralist and veterinary perception was always an external perspective of humans on animals.

Multilingualism also plays a major role in terms of research ethics, since knowledge is not only transferred through communication but can also be acquired in the writing process in epistemic writing, whereby knowledge is generated through the writing process itself. This happens in different phases of a project, for example during data analysis [41]. In epistemic writing in a foreign language, the highest level of epistemic writing is not reached. As less knowledge is thus acquired, all non-native speakers are disadvantaged in their knowledge acquisition, which can lead to epistemicide, a reduction of acquisition and production of new knowledge and transferred knowledge within the project. Is it ethically correct to deny some project participants the possibility of optimal knowledge acquisition through the use of a lingua franca? The question of who gets to decide on the language(s) to be used in a project is highly relevant to research ethics [43]. Beyond the mere issue of language use, the domain of ethics in these pluri-epistemic settings eventually reaches aspects of the hierarchies of values guiding health interventions, or what might constitute ‘acceptable’ trade-offs in human–animal–ecosystem health. As an example, a recent publication on an indigenous view of the determinants of planetary health has proposed the use of new categories such as ‘ancestral legal personhood designation of elements’ to afford legal rights to the health of entities such as rivers, lakes, or mountains [44]. This is, however, a topic not yet ripe for discussion in most academic settings.

Collaboration for One Health is not feasible without multilingualism. Even in projects conducted exclusively with speakers of a single language, intralingual multilingualism occurs through interaction with domains of practice. Multilingualism is an essential component of transdisciplinary and fair cooperation and must be addressed and clarified at the beginning of the project. Most recently, a project on hygiene and antibiotic resistance in the poultry production chain in Palestine in collaboration with Birzeit University used the experience of intercultural, multilingual transdisciplinary processes as a basis for equal collaboration in a One Health approach.

## 17. One Health Methods

If we want to study the health of humans, animals, plants and, figuratively speaking, the environment at the same time, we need new methods to do so. The interdependencies can be investigated with statistical methods, for example. The dependency of one sector on the other is investigated in a regressive way. For example, we studied the dependence of human brucellosis seroprevalence on seroprevalence in sheep, goats, and cattle in Kyrgyzstan. In one run, we were able to identify the source of transmission through sheep. We thus gained an added value of knowledge compared to studies that only examined humans or animals and therefore could not prove such a link [45]. If we want to study the impact of interventions such as mass vaccination, we need to resort to mathematical models that dynamically study the interfaces between humans, animals, and the environment, as shown in the example of brucellosis control in Mongolia, described above (Equation (1)) [17]. These methodological approaches can be further extended to the interaction of human health and nutrition in the context of crop production, ecosystem biodiversity, and climate change impacts. In this way, we extend the original One Health idea, focused on the collaboration of human and veterinary medicine, to the whole human–environmental system, or the social-ecological system (SES). This raises questions about natural resources.

## 18. Towards a One Health Game Theory

We take up the concepts of Elinor Ostrom, the pioneer of new institutional economics and management of the commons. We explain the health of humans, animals, and plants as the result of the complex interrelationships within an SES as Health in Socio-Ecological Systems (HSES) [46]. This includes also sustaining ecosystem services such as clean water and air or pollination. For example, Q’eqchi’ people in Guatemala include Kawilal Tzultaqa’ in their concept of health, which means the health of water, of soil, and of air. up to the health of the entire ecosystem. The underlying processes are complex, multivariable, non-linear, stratified, and changing. For example, with the use of artificial fertilizers and pesticides, we achieve large harvests that guarantee food security. On the other hand, the use of pesticides has negative impacts on human and animal health and leads to loss of biodiversity. Loss of biodiversity in turn leads to a higher risk of diseases. Integrated approaches to health, such as One Health, are useful in this regard, but need to be further developed. We are inspired by Elinor Ostrom’s game-theoretical approaches to management of the commons [47]. We further owe the connection with the principle of cooperation to Maria Zinsstag, who drew attention to the book Evolution, Games and God: The principle of Cooperation by Nowak and Coakley [48].

The health of humans and animals is multifaceted. It has a purely private dimension and is arguably the highest private asset in our life. Our own health is thus our most valuable private good. However, health also clearly has an important public and common dimension. Through being infected by another person or infecting another person with a preventable (or non-preventable) disease, health becomes eminently public and global. We can consider freedom of disease in its non-rivalrous and non-excludable quality [49] as a common good in Ostrom’s understanding [47].

By analogy, the unhindered spread of diseases, leading to outbreaks or endemic stable transmission of disease can be considered as a “tragedy of the commons” in Hardin’s sense [50]. For example, ongoing transmission of rabies in many West and Central African countries is indeed a tragedy, causing the deaths of tens of thousands of people, mostly children, every year [51]. In contrast, if all people exposed to rabies-suspected animals could be protected with post-exposure prophylaxis, human deaths could be avoided. If successful dog mass vaccination campaigns at sufficient coverage would be conducted, rabies could even be eliminated [52], and its cumulative cost would be the lowest [53]. Such a high level of cooperation across the levels of social organization, from household to national governments, which is needed to eliminate dog-mediated rabies, requires transdisciplinary participatory process between all actors of civil society, authorities, and academic actors.

We insert our initial thinking of One Health as an added value of a closer cooperation between human and animal health into the broader game theory of cooperation at the level of evolution, economics, and natural resource management. We consider the health of humans and animals, and metaphorically of the ecosystem, as a basic social good in Rawls’ vocabulary [54].

However, the premises of game theory have to be carefully assessed both for their use at the individual (individual health) level and as collective action for public health [55]. Cooperation for collective One Health action, using transdisciplinary participatory processes at different levels from individuals and households to communities, provinces, countries and the international level, becomes a central feature of the One Health approach. The different scales of social layers match well with the multilayered social resilience concept for mitigation research [56]. We currently develop One Health game theoretical frameworks by specifying utility functions between the different public and animal health actors at different levels (individual and at different levels of collective action), as a theoretically well-grounded methodological approach to solving health problems in social-ecological systems [30]. To understand human–animal–environment relationships, we need to know their cultural, religious, social, and psychological foundations in a particular context. One Health approaches depend on the closest possible collaboration with cultural and social sciences. This not only contributes qualitative variables to epidemiological research, but creates entirely new scientific approaches that critically examine an anthropocentric view through a broader view of humans and other species in their environment. For example, Donna Harway’s proposal in “Chthulucene” to rename “Humanities” to “Humusities” is noteworthy [57]. 

## 19. Operationalising One Health

One Health approaches develop scientific principles for solving complex health problems in human–environment systems. These only have a broad impact if they are implemented at different levels of society. In Switzerland, Andrea Meisser contributed to the operationalisation of One Health and analysed the processes within the highly decentralised administration [58]. He mediated the cooperation with the cantons of Ticino and Basel, where Switzerland’s first cantonal One Health policy was created under the leadership of Anne Lévy (Chapter 14) [29]. In 2017, the so-called One Health sub-body emerged (Unterorgan “One Health” Decree 1 November 2017), which organised four Swiss federal offices to work together based on the definition of the Swiss TPH. Marcel Tanner established contact with federal parliamentarians, who submitted a postulate to the Federal Council to promote integrated approaches to antibiotic resistance research.

At the international level, we investigated operationalisation of One Health at the request of the Chatham House think tank in London. Country reports to the WHO under the International Health Legislation (IHR 2008) and to the World Organisation for Animal Health (WOAH) within the Performance of Veterinary Services (PVS) reports were examined for mention of One Health approaches.

Examples from joint health services and infrastructure, surveillance-response, antimicrobial resistance surveillance, food safety and food security, environmental hazards, water and sanitation, and zoonoses control clearly demonstrate incremental benefits of One Health approaches. One Health approaches appear to be most effective and sustainable in the prevention, preparedness, and early detection/investigation of evolving risks/hazards, and the evidence base for their application is strongest in the control of endemic and neglected tropical diseases. For the benefits to be maximised and extended, improved One Health Operationalisation is needed with the strengthening of multisectoral coordination mechanisms at global, regional, and national levels [6]. There remain still major barriers to the leverage of One Health in the medical sector, narrow government visions, and the sheer lack of willingness to cooperate.

With the extension of the Tripartite Agreement on the One Health cooperation of WHO, FAO, and WOHA (founded as OIE) to a “Quadripartite” by including the United Nations Environment Programme (UNEP), the environmental sector was formally included at the international level. We are currently advising UNEP on how to build One Health in the environmental sector.

## 20. Conclusions and Outlook

We can summarize that the history of ideas and processes leading to the development of One Health research at the Swiss TPH were inspired by far-sighted and open ideas of the directors and heads of departments, without exerting too much influence. They followed the progressing work and supported it with further ideas. These in turn were taken up and further developed by a growing number of individual scientists. These ideas were related to other strands of knowledge from economics, molecular biology, anthropology, sociology, theology, and linguistics. We endeavour to relate Western biomedical forms of knowledge generation with other forms, such as Mayan medicine. One Health, in its present form, has been influenced by African mobile pastoralists’ integrated thinking that have been taken up into Western epistemologies. The intercultural nature of global and regional One Health approaches will inevitably undergo further scrutiny of successful ways fostering inter-epistemic, transdisciplinary interaction. In this way, there is no need for a “One Health of peripheries” as proposed by Baquero et al., [59] there is just One Health.

Today, we see a broader social potential of One Health as an integrated approach. One Health demonstrates the added value of cross-sectoral inter- and transdisciplinary collaboration. Our many years of experience with participatory transdisciplinary processes and our participation in the Transdisciplinarity Network (TD-net, www.transdiscsiplinarity.ch, accessed on 9 August 2022) of the Swiss Academies contributed significantly to the development of the policy paper of the Organization for Economic Co-operation and Development. This places participatory transdisciplinary cooperation for problem solving in society as a whole in a larger societal context and goes far beyond health issues [60].

Now theoretically well grounded, the One Health approach of seeking benefits for all through better and more equitable cooperation can clearly be applied to engagement in solving major societal problems such as social inequality, environmental protection, climate change mitigation, biodiversity conservation and conflict transformation. More and more scientific circles are advocating extinction rebellion or revolts to combat biodiversity loss or climate change [57]. However, revolts would inevitably lead to violence and delays in the implementation of urgently needed improvements of social equity, animal protection and welfare, halting of biodiversity loss and environmental protection. A worldwide consensual search for cooperation, as demonstrated through participatory One Health research, seems far more effective for improving social and ecological conditions.

## Data Availability

Not applicable.

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
