# Peer review of "The Promotion and Development of One Health at Swiss TPH and Its Greater Potential"

_diseases, 2022, doi:10.3390/diseases10030065_

Round 1

Reviewer 1 Report

Dear authors,

a minor grammatical issue or two in the text e.g. line 83 should be "look" not "looks" and sentence lines 126-128 - was it the veterinary expertise that was familiar with brucellosis or Jakob Zinsstag.

The formula line 160 needs the symbols to be explained - otherwise not understandable by the general reader

Line 250 It is the University of Queensland in Brisbane  NOT the University of Brisbane.

Line 251 - may be mentioned Joint international courses - these were held between STPH and UQ

Not needed for the paper - but for future consideration: it would be useful to develop an example of the Game theory concept for a non-communicable disease - given the broadening ( necessary) of the One Health concept as described beyond zoonotic infectious diseases.

Not needed for the paper but for future consideration - is the attempt to use game theory - a partially reductionist approach - even with the broadening of disciplinary "data" and understanding  into this.

Really enjoyed the paper. And a great tribute to Marcel Tanner.

Author Response

a minor grammatical issue or two in the text e.g. line 83 should be "look" not "looks" and sentence lines 126-128 - was it the veterinary expertise that was familiar with brucellosis or Jakob Zinsstag.

Answer: The word look is corrected and the sentence about the veterinary expertise on brucellosis of Jakob Zinsstag is re-formulated.

The formula line 160 needs the symbols to be explained - otherwise not understandable by the general reader

Answer: The symbols in the formula are all explained now.

Line 250 It is the University of Queensland in Brisbane  NOT the University of Brisbane.

Answer: University of Brisbane is replaced with Unviersity of Queensland in Brisbane

Line 251 - may be mentioned Joint international courses - these were held between STPH and UQ

Answer: The joint courses between UQ and Swiss TPH are mentioned now.

We thank the reviewer for the careful reading and the additional comments.

Reviewer 2 Report

I reviewed the manuscript entitled “The promotion and development of One Health at Swiss TPH and its greater potential. In this manuscript authors explain and present the perspectives of the One Health concept.

Overall, I found this document as an interesting essay. I consider that the topic covered in this manuscript is very important for the promotion of health.  I think the idea of integrating several disciplines in one concept (health) is a key factor for success.

This are some of my suggestions to improve the quality of this manuscript:

-          I suggest the authors to improve the content of the introduction. I think is necessary to include more information about how social inequality is currently affecting public health.  Are there other markers like corruption, government effectiveness that should be considered to explain social inequality and so that are affecting public health?

-          Another suggestion would be to include a figure summarizing the overall concept

-          I suggest putting in perspective other potential diseases that can be considered as a part of this concept.  

-          For the conclusion, what are the main challenges to promote the success of this concept?

Author Response

This are some of my suggestions to improve the quality of this manuscript:

-          I suggest the authors to improve the content of the introduction. I think is necessary to include more information about how social inequality is currently affecting public health.  Are there other markers like corruption, government effectiveness that should be considered to explain social inequality and so that are affecting public health?

Answer: We agree with the reviewer and have added a sentence emphasizing the social aspects besides the ecological and climate issues.

-          Another suggestion would be to include a figure summarizing the overall concept

Answer: We agree with the author. One Health, as we describe it here is so multifaceted that we would prefer an artistic interpretation of the concept rather than a schematic drawing. Unfortunately, we don’t have an artist at hand at such short notice. We thank the reviwer and take up the suggestion for a forthcoming publication.

-          I suggest putting in perspective other potential diseases that can be considered as a part of this concept. 

Answer: We refer at numerous places to the different editions of our textbook which contains an extended set of other diseases. We hope the readers will consult the textbook.

-          For the conclusion, what are the main challenges to promote the success of this concept?

Answer: We agree with the reviewer and have added a sentence in the section on the operationalization of One Health.

We thank the reviewer for the valuable comments.

Reviewer 3 Report

This is a review document of the Swiss TPH strategy for development of a One Health Strategy by Jakob Zinsstag, and covers the development of the ideas in one health, under the period of the leadership of Professor Tanner, and beyond

Minor comment: the authorship does seem to be dominated by the Europeans here, and could be viewed by some as being somewhat colonialist.  Consider placing the investigators from outside of Switzerland in more prominent positions (perhaps 50% of the first and last three authors).  This is quite an important point since the authors are talking about social inequality.  It is good to see the Effectiveness formula in such a high position – if our colleagues could understand this simply equation then things would move along more simply!

The paper is well written, and easy to understand – and an enjoyable read as a review.  I have no other useful comments to add.

Author Response

Minor comment: the authorship does seem to be dominated by the Europeans here, and could be viewed by some as being somewhat colonialist.  Consider placing the investigators from outside of Switzerland in more prominent positions (perhaps 50% of the first and last three authors).  This is quite an important point since the authors are talking about social inequality.  It is good to see the Effectiveness formula in such a high position – if our colleagues could understand this simply equation then things would move along more simply!

Answer: This is a scientific paper in which the authors are listed according to their contribution. The authors also consented to their position in the list of authors. There is nothing colonial about this paper because there are more European authors. In contrast re-arranging the list for geographical representation and not for scientific merit would be ethically problematic. We kindly request to leave the author list as it stands. We thank the reviewer for the comment on the effectiveness formula.